# Developing a More Reliable Aerial Photography-Based Method for Acquiring Freeway Traffic Data

Chi Zhang [1,2], Zhongze Tang [1,2], Min Zhang [3,*], Bo Wang [1,2] and Lei Hou [4]

1  School of Highway, Chang'an University, Xi'an 710064, China; zhangchi@chd.edu.cn (C.Z.); 2020221312@chd.edu.cn (Z.T.); wb1010110wb@chd.edu.cn (B.W.)
2  Engineering Research Center of Highway Infrastructure Digitalization, Ministry of Education, Xi'an 710000, China
3  College of Transportation Engineering, Chang'an University, Xi'an 710064, China
4  School of Engineering, RMIT University, Melbourne, VIC 3000, Australia; lei.hou@rmit.edu.au
*  Correspondence: minzhang@chd.edu.cn; Tel.: +86-139-9128-6571

**Abstract:** Due to the widespread use of unmanned aerial vehicles (UAVs) in remote sensing, there are fully developed techniques for extracting vehicle speed and trajectory data from aerial video, using either a traditional method based on optical features or a deep learning method; however, there are few papers that discuss how to solve the issue of video shaking, and existing vehicle data are rarely linked to lane lines. To address the deficiencies in current research, in this study, we formulated a more reliable method for real traffic data acquisition that outperforms the traditional methods in terms of data accuracy and integrity. First, this method implements the scale-invariant feature transform (SIFT) algorithm to detect, describe, and match local features acquired from high-altitude fixed-point aerial photographs. Second, it applies "you only look once" version 5 (YOLOv5) and deep simple online and real-time tracking (DeepSORT) to detect and track moving vehicles. Next, it leverages the developed Python program to acquire data on vehicle speed and distance (to the marked reference line). The results show that this method achieved over 95% accuracy in speed detection and less than 20 cm tolerance in vehicle trajectory mapping. This method also addresses common problems involving the lack of quality aerial photographic data and accuracy in lane line recognition. Finally, this approach can be used to establish a Frenet coordinate system, which can further decipher driving behaviors and road traffic safety.

**Keywords:** remote sensing applications; traffic data acquisition; UAVs; SIFT; YOLOv5; DeepSORT; Frenet coordinate system

## 1. Introduction

The question of how to increase highway driving safety is a major issue around the globe. In 2021, the Decade of Action for Road Safety campaign, launched by the World Health Organization (WHO), set a goal that by 2030, global road traffic fatalities should be reduced by 50% [1]. A thorough understanding of driving data is required to improve road safety. Numerous studies have shown that the driving behaviors of drivers on the same road are correlated; therefore, speed and trajectory data can be utilized to quantify the risk of driving accidents [2–4]. The traditional methods used for collecting vehicle speed and trajectory data primarily use driving simulations, cross-section speed measurements, and real driving on the road; however, the data obtained using the traditional methods can lack accuracy and continuity, and therefore, they cannot be used to measure data correlations. In recent years, with the wide application of unmanned aerial vehicles (UAVs) in remote sensing (RS), driven by their academic and commercial success [5], data acquisition methods based on surveillance and aerial video have become more popular [6–8]. Even though these methods have promoted the application of remote sensing technology, such as finer

object detection and real-time tracking [9–11], they have yet to fully address problems with changing illumination, object occlusion/obstruction, and environmental noise.

Acquiring driving data based on video footage has been well researched, and a wide range of methods have been formulated, including the inter-frame difference, background difference, and optical flow methods; however, these methods are limited by their identification accuracy and vulnerability to environmental noise [12,13]. With the development of neural networks, such as the convolutional neural network (CNN) and the recurrent neural network (RNN), object detection speed and accuracy have improved significantly [12,14]. The "you only look once" (YOLO) model, adapted from the Darknet network, is proven to be effective for UAV-based object detection, and thus was used in this study. It is worth noting that the process of improving driving safety is multi-faceted. In addition to the vehicle data itself, the correlation between the road and the vehicle, such as the relative position of lane and vehicle, can also be used to examine road safety. Current commonly used lane line recognition algorithms, such as edge detection and deep learning-based feature detection, are likely to fail when an object to be detected lacks clarity.

To achieve extraction of real vehicle data from aerial video with higher precision and better reliability, it is imperative to conduct an in-depth study in three areas: vehicle identification, vehicle tracking, and vehicle and lane line correlation. Section 2 examines the related research in these fields; Section 3 presents the settings of our traffic flow video capture technique; Section 4 elucidates the steps of acquiring high-precision real vehicle data and the data processing method; Section 5 describes the data results associated with the selected road sections; and Section 6 provides a discussion and conclusion.

## 2. Literature Review

Roads will have a variety of effects on traffic safety. Aside from the road alignment, which affects the driver's judgment, markings, pavement materials, and damage all have an impact on the vehicle's driving condition [15,16]. These factors are ultimately described by the vehicle speed and trajectory, upon which we can base a comprehensive road safety evaluation. These data can be obtained by different techniques, e.g., predicting vehicle speed, simulating driving, measuring cross-section speed, observing real driving behavior, examining trajectories, and so on [3,17]. The common problems of these techniques include data authenticity and acquisition discontinuity [18]. Nowadays, a trending technique is analyzing the natural flow data (e.g., NGSIM, HighD, etc.) from driving footage acquired from CCTV surveillance cameras, drone cameras, and other photographic devices. This technique is limited as well, as it can only reflect the driving situation of a certain road section.

DADS was the first commercialized road driving simulation system, developed at the end of the 19th century, but it can only run on UNIX and has few functions [19]. As computer-aided design, geographic information systems (GIS), and building information modelling (BIM) technologies continue to evolve, a series of driving simulation systems, such as UC-win/road, CarSim, TruckSim, and CARRS-Q, have been developed to support a variety of road safety and driving behavior studies [20–23]. The rationale is that they can generate a wide range of simulated driving scenarios for testing and analysis, which can help address real-world driving challenges to some extent. The development of radar and sensor technology has further facilitated the acquisition and accuracy of vehicle information, such as real-time position, speed, and braking behavior [24,25]; however, this method also has limitations. First, it is not robust enough to examine the driving behavior of many vehicles on the same road. Second, it is based on cross-section speed measurements, and this technique cannot handle vehicle occlusion problems [26–28]. Emerging UAV equipment and image processing algorithms enable a novel approach for vehicle data collection and analysis using high-altitude aerial photography. NGSIM and HighD are representative datasets generated from this technology [7,8]. These datasets have high precision and are rich in traffic flow data [29–31]; however, there are still limitations. For instance, the collected road sections in these databases have a few bifurcate structures and

horizonal/vertical geometry, which cannot sufficiently support studies investigating car following and lane changing.

More literature on vehicle speed and trajectory data is summarized in Table 1. These data can play an important role in road safety and driving behavior research. Lili et al. [32] reviewed the traffic flow research and noted that the scarcity of real trajectory data is a key issue. Vehicle trajectory data with high precision and validity can be quite beneficial, but the acquisition, management, and application of such data are huge challenges [33]. The distance information between vehicle and lane line is another important indicator to understand driving safety and behavior, but the current research has not fully taken this into consideration; therefore, it is argued that obtaining reliable and precise data in this regard is another major issue that could benefit from the use of UAVs. There are two types of video-based vehicle recognition methods: one is based on optical properties, such as inter-frame difference, background difference, and optical flow, and the other is based on deep learning object detection algorithms. For example, the YOLO algorithm proposed by Redmon et al. [34] is widely used in the field of target recognition, and is also applied in this paper.

**Table 1.** Summary of vehicle speed and trajectory data studies.

| Literature | Data Source | Conclusions and Results |
|:---:|:---:|:---:|
| [3] | Various speed profiles on 153 Dutch expressway curves | Deflection angle and curve length are connected to curve speed. Regardless of horizontal radius or speed, vehicles come to a halt and decelerate roughly 135 m into the curve. |
| [35] | NGSIM dataset | Examines elements that influence likelihood of collision during lane changes from the perspective of vehicle groups, as well as unobserved heterogeneity of individual lane change movements. |
| [36] | HighD dataset | Active safety management strategy developed by combining traffic conditions and conflicts. |
| [32] | Review | Use of traffic flow research based on trajectory data from microscopic, mesoscopic, and macroscopic perspectives is reviewed, with paucity of data at this stage identified as a major issue. |
| [37] | Combines road collection with driving simulation data | Using trajectory data, quantifies back-end risk and proposes thresholds. |
| [38] | Less accurate quasi-vehicle trajectory data | Bayesian matching case control logistic regression model is established to explore impact of traffic parameters along quasi-trajectory of vehicles on real-time collision risk. |
| [39] | Second Strategic Highway Research Program Naturalistic Driving Study (SHRP2 NDS) | Understanding and quantifying effects of factors such as road speed, driver age and gender, vehicle class, and location on longitudinal and lateral acceleration cycle rates. |
| [40] | empirical vehicle trajectory data collected from Interstate 80 in California, USA, and Yingtian Expressway in Nanjing, China | According to the three-phase theory, three regression models were created to quantify impact of traffic flow factors and traffic states on collision risk, allowing the evaluation of collision risk for distinct traffic phases and phase transitions. |
| [41] | Driving simulator | Investigates efficiency of static and dynamic merging control in management of urban and rural expressway traffic. |
| [42] | HighD | Collision prediction using a random parameter logistic regression model that takes into account data heterogeneity problem. |
| [43] | Smartphone GPS trajectory data | Investigates behavior of vehicles when faced with traffic congestion and road closures. |

The inter-frame difference method makes use of the strong correlation between single-frame and multi-frame intervals in the image sequence to compare the grayscale difference and threshold value of pixels, from which moving objects can be identified [44,45]. When vehicles are moving fast, this method would suffer from the so-called ghosting problem,

which is mistakenly identifying vehicles as background objects. The background difference method leverages a constructed static background model to reflect a foreground moving object [46,47]. This method cannot continuously track a vehicle's coordinates. The optical flow method can calculate the traveling speed and direction of a target through pixel intensity change and correlation analysis [48,49]; however, it is less reliable when illumination conditions change. An accurate vehicle trajectory extraction method using the Canny ensemble detector and kernelized correlation filter was developed [50]; however, this method tends to have data corruption problems, especially when applying vehicle trajectory tracking. The deep learning target instance detection approach can effectively solve some of the abovementioned problems, and it makes use of trained vehicle classifiers to recognize different vehicle types from consecutive frames. In a tough environment, this method can achieve a better performance [12,14].

Obtaining the speed and trajectory data of all cars throughout an entire road segment is more favorable for traffic safety analysis. This research presents a reliable method to obtain speed and trajectory data that addresses some shortcomings of traditional methods. This method leverages UAVs to acquire real traffic videos. It is a valid and accurate method that can relate vehicle data to road information. The road alignment in this study varies, and includes straight roads, sharp turns, ramps, intersections, road exits, and small clear distance roads located in Sichuan and Shaanxi, China.

In this paper, we aim to develop a more reliable method to extract vehicle data. In order to obtain better data, the aerial photography environment is a controllable condition, so UAV video extraction is performed in an ideal environment regardless of the conditions, such as strong wind, rainfall, or low visibility, which would adversely affect drone aerial photography. We collected a huge volume of streaming high-definition traffic video data to create a training set for YOLOv5 and Darknet. Based on the evaluation, the target frame was more stable than the optical approaches. YOLO has an inferior detection effect when target objects are relatively small and close; however, this would not be the case in the free-flow condition of highway vehicle detection. It should be noted that the robustness of the algorithm in terms of lane line recognition is affected by lighting, reflection, noise, and other factors. To ensure the accuracy of vehicle distance data, in this study, we propose calibrating the lane line parameters at the beginning. An integration of video registration, vehicle recognition, and Python big data processing underpins the vehicle data extraction method of this study. The vehicle recognition rate and data extraction accuracy in different environments show that the developed method outperforms the traditional methods in terms of data integrity and accuracy. Finally, in this study, we also examine noise data and construct a Frenet coordinate system to display the data patterns in different environments.

## 3. Collection of Real Traffic Flow

### 3.1. Research Location

In order to verify the effectiveness of data extraction in different environments, a number of mountainous highways in Shaanxi and Sichuan, China, were selected. The data were collected in different road and traffic conditions (Table 2 and Figure 1).

### 3.2. Research Equipment

UAVs can be used in many applications, such as wide-area traffic monitoring and management, traffic parameter gathering, and so on. According to the study's data requirements, it was ultimately determined that the data would be collected using UAV fixed-point aerial photography. The drone used in this field investigation was a DJI Mavic 2 Zoom; it uses six batteries, and each battery can last 25 min. The built-in 12 million pixel lens of the UAV can support steady video recording of 4 k + 60 fps. This UAV is equipped with a 3-axis mechanical gimbal (pitch, roll, and pan) (Figure 2) with a pitch range of $-90°$ to $+17°$ (extended).

### 3.3. Research Process and Data Acquisition

The aerial photography was done vertically (−90°) to eliminate calibration errors due to the oblique view of the angle at different positions. A comparative experiment on the effectiveness of videos at various heights was conducted, and the findings are shown in Table 3. The drone height was kept in between 210 and 230 m and the photographed road range was about 300 and 160 m wide. To eliminate shooting angle errors, a fixed-point vertical aerial shooting approach was used.

**Table 2.** Road profiles.

| Province | Highway | Road section | Features |
|----------|---------|--------------|----------|
| Shaanxi | Baomao expressway | General main line segment | Free-flow straight sections, with stable and fast speed, and a small amount of lane-changing behavior. |
| Sichuan | Yakang expressway | Tianquan interchange expressway exit | A number of lane-changing behaviors on the main line, which has a big radius and a right-biased curve. |
| Sichuan | Yakang expressway | Duogong interchange ring ramp | Vehicle direction and speed fluctuate dramatically in the circular curve section with a small radius. |
| Shaanxi | Jingkun expressway | Huangguan tunnel small net distance exit | Main line is straight with a lane change and clear acceleration and deceleration characteristics. |
| Sichuan | Yakang expressway | Luding interchange ring ramp | Heavy traffic, slow speed, and many interfering vehicles. |

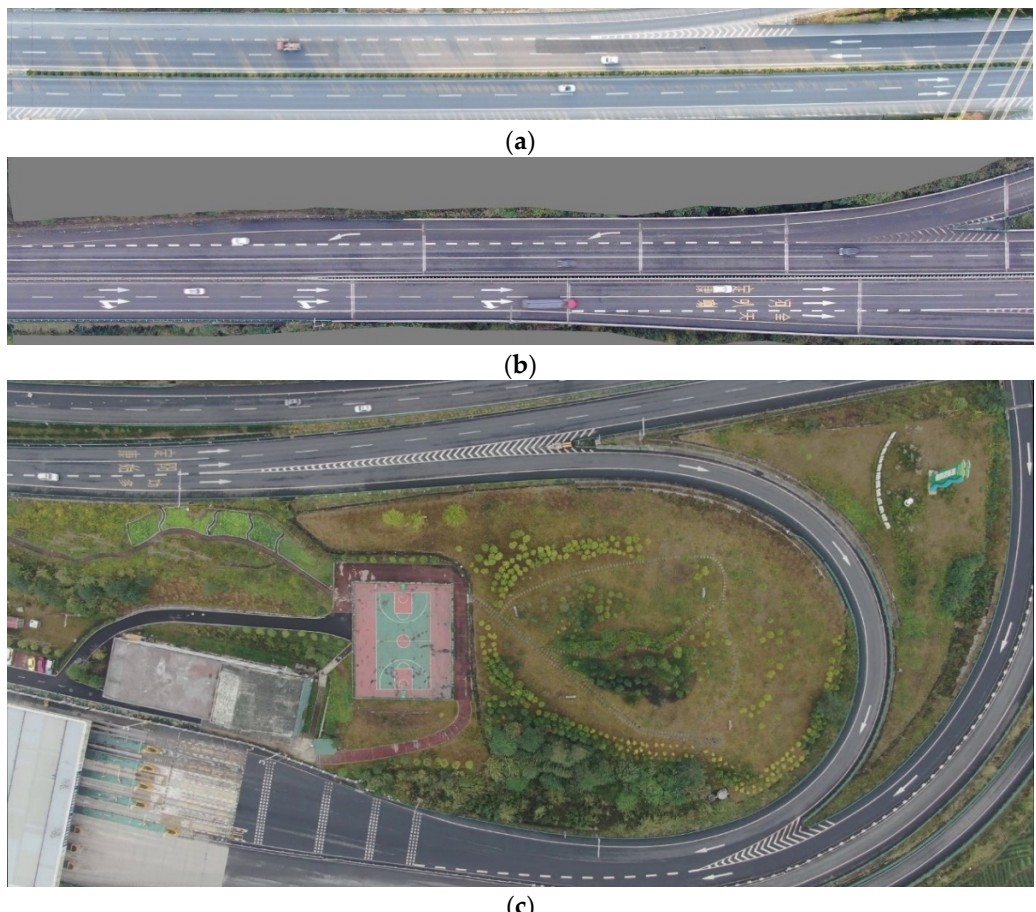

(**a**)

(**b**)

(**c**)

**Figure 1.** *Cont.*

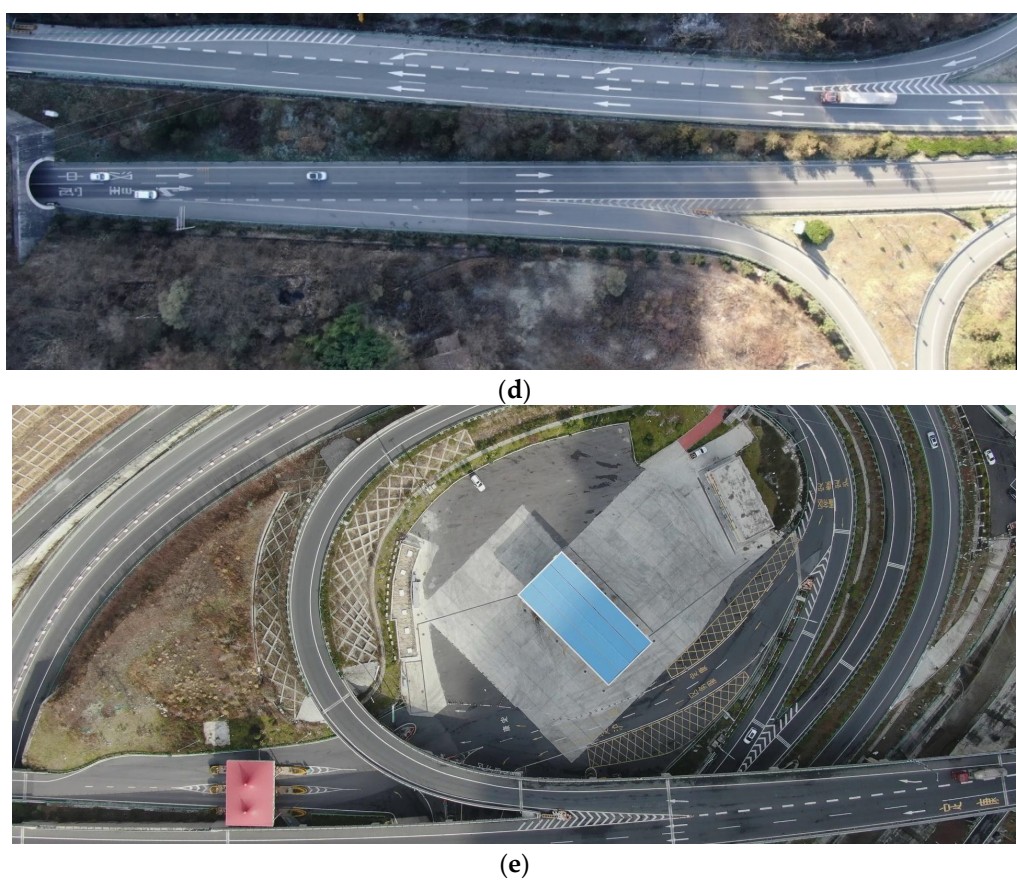

(**d**)

(**e**)

**Figure 1.** Research locations: (**a**) main section of Baomao expressway; (**b**) Tianquan interchange exit of Yakang expressway; (**c**) Duogong interchange ring ramp of Yakang expressway; (**d**) small net distance section of Huangguan tunnel of Beijing–Kunming expressway; (**e**) Luding interchange ring ramp of Yakang expressway.

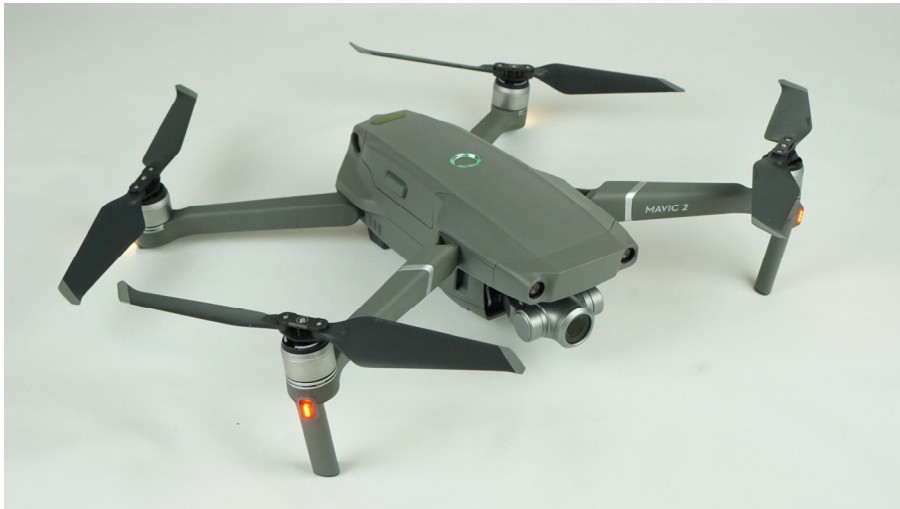

**Figure 2.** DJI-Mavic 2 Zoom.

**Table 3.** Flight altitude analysis.

| Height (h) | Effect of Vehicle Recognition | Error Analysis |
|---|---|---|
| <200 m | 95% recognition, continuous tracking | Road section's shooting range is narrow, vehicle's process time entering and leaving the screen is long, and the process speed error is considerable, resulting in more erroneous data during entry and exit process. |
| >250 m | 93% recognition, some vehicles have identification (ID) switching | When faced with a more complex road environment, the vehicle is too small at this height, and it is easy to cause ID switch when the white car and white marking line overlap; if the height is too high, wind speed is high, and drone stability is poor, the real distance represented by one pixel is large, and the system error increases. |
| 200–250 m | 98% recognition, continuous tracking | At this height, it can be adjusted according to road environment and wind speed to achieve continuous and stable tracking of the vehicle, with tiny errors between vehicle speed and trajectory. |

## 4. Materials and Methods

The method outlined in this paper is divided into 5 steps, as shown in Figure 3:

(1) Image registration: In drone aerial photography, camera shake is unavoidable and will result in significant mistakes. To solve this problem, we applied the SIFT algorithm to register the video.

(2) Target detection: Following a comparative analysis, we employed YOLOv5 based on deep learning to realize vehicle detection, calibrating over 6000 images of cars of various forms and obtaining the best recognition model after 100 training cycles.

(3) Continuous vehicle tracking: After achieving high-precision vehicle identification, the DeepSORT algorithm was used for continuous vehicle tracking and trajectory extraction, and the vehicle speed was recovered using the distance calibration value and time interval.

(4) Lane line calibration: To ensure a solid linkage of vehicle data with lane lines, we first calibrated lane lines and then computed the relationship with the vehicle.

(5) Data extraction: We used Python, based on the Helen formula, to extract vehicle and lane line data, and established a Frenet coordinate system for data display.

### 4.1. Image Registration Based on SIFT Algorithm

Both wind and the machine itself can cause the drone to wobble slightly, which can affect the video shooting quality (Figure 4a,b). In our study, this problem leads to certain errors in the extraction of vehicle trajectories (which will be discussed in greater depth later) but it does not affect vehicle speed extraction.

To tackle the wobbling issue, in this study, we applied the SIFT registration algorithm to ensure that the relative position of vehicles to the road did not change too much over time. The SIFT algorithm has been widely used in image recognition, image retrieval, and 3D reconstruction [51]. Unlike conventional target detection algorithms, which are very sensitive to image size and rotation, SIFT is not sensitive to image size and orientation, and has the capability to resist illumination change and environmental noise.

We set the reference image as $f(x,y)$ and the registered image as $g(x,y)$. If point $(\hat{x}, \hat{y})$ on the reference image corresponds to point $(x,y)$ in the image to be registered, an affine relationship exists:

$$\begin{bmatrix} \hat{x} \\ \hat{y} \end{bmatrix} = k \begin{bmatrix} \cos\theta & \sin\theta \\ -\sin\theta & \cos\theta \end{bmatrix} \cdot \begin{bmatrix} x \\ y \end{bmatrix} + \begin{bmatrix} \Delta x \\ \Delta y \end{bmatrix} \tag{1}$$

where $k$ is the scale parameter, $\theta$ is the rotation angle, and $\Delta x$ and $\Delta y$ are the translations of the two axes, respectively.

The road can be matched and the images of all frames in the video can be corrected to the same position as the first frame using more than 4 feature points, according to the

SIFT algorithm. Figure 5 shows that the changes in relative position induced by the camera shake of the same vehicle over time will no longer exist due to the registered video. Cars between distinct frames can be identified as fixed elements by SIFT, especially vehicles with slower speeds, resulting in changes in the vehicle's driving state due to registration. This problem was solved in this study by selecting the region of interest in the non-lane range when registering. The registration has no effect on the vehicle state in real operating settings, according to the before and after comparison experiment.

To show the effect of the registration more intuitively, a straight road portion was used for comparison, and 2 frames of photographs taken at different times were superimposed to evaluate the difference, as illustrated in Figure 6.

From Figure 6a, it can be seen that there is an offset between the images at the same position on the road before registration. It can be seen in Figure 6b that the image has a black border to cancel the shaking during registration, so that the image is consistent with the first frame, ensuring that the relative position of the vehicle and the lane line will not change with the shaking camera. The black borders formed by the registration are acceptable and have no effect on subsequent data gathering because the shaking is minimal.

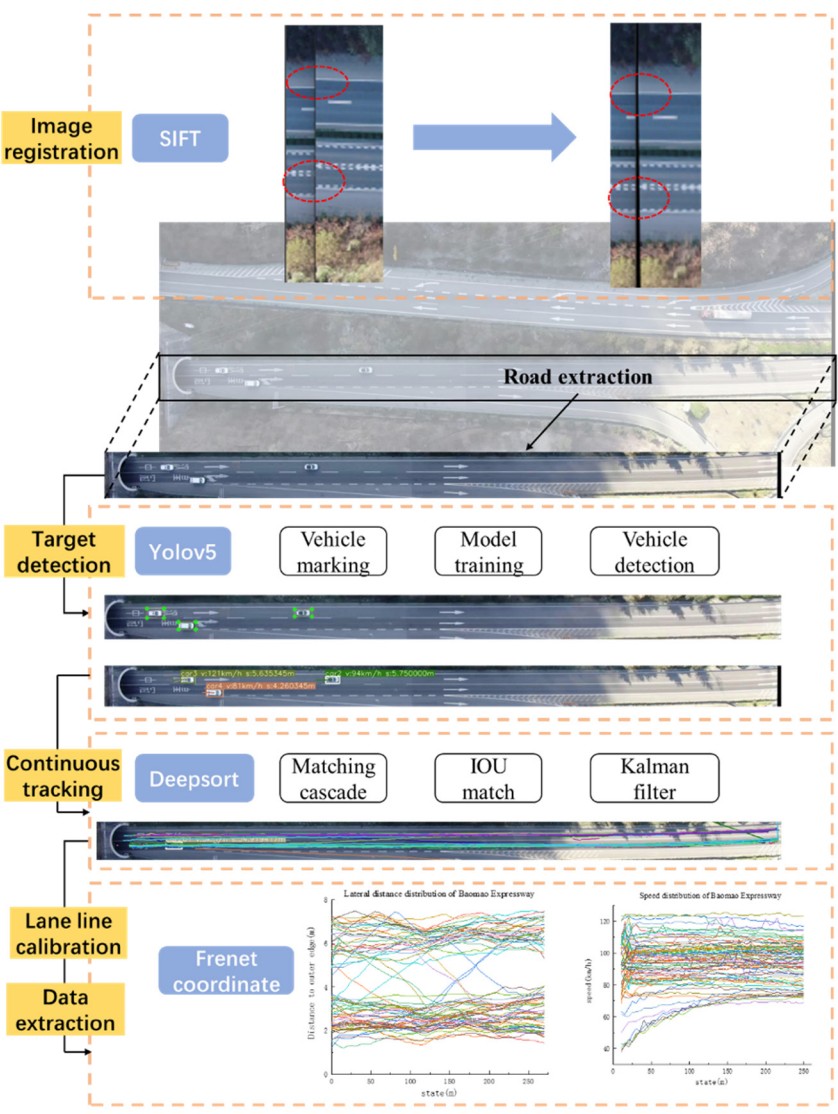

**Figure 3.** Data extraction process.

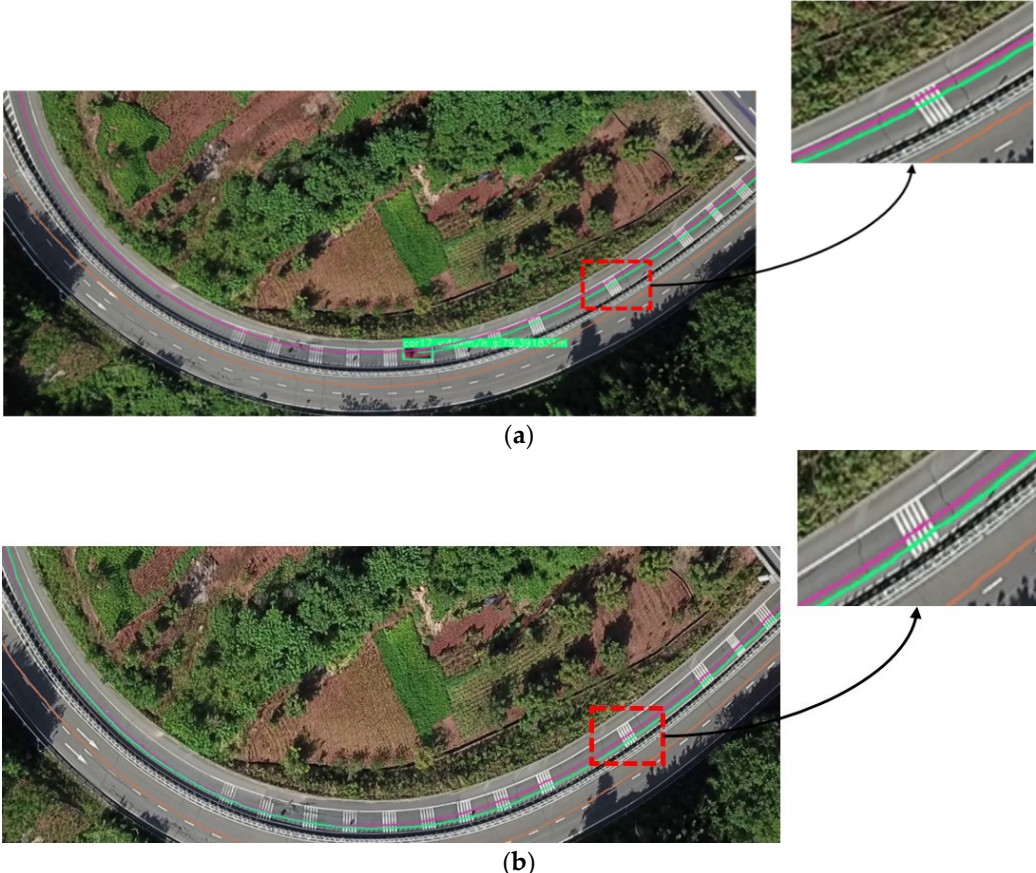

**Figure 4.** Camera wobbling issue: (**a**) vehicle trajectory at previous frame; (**b**) vehicle trajectory at a later frame.

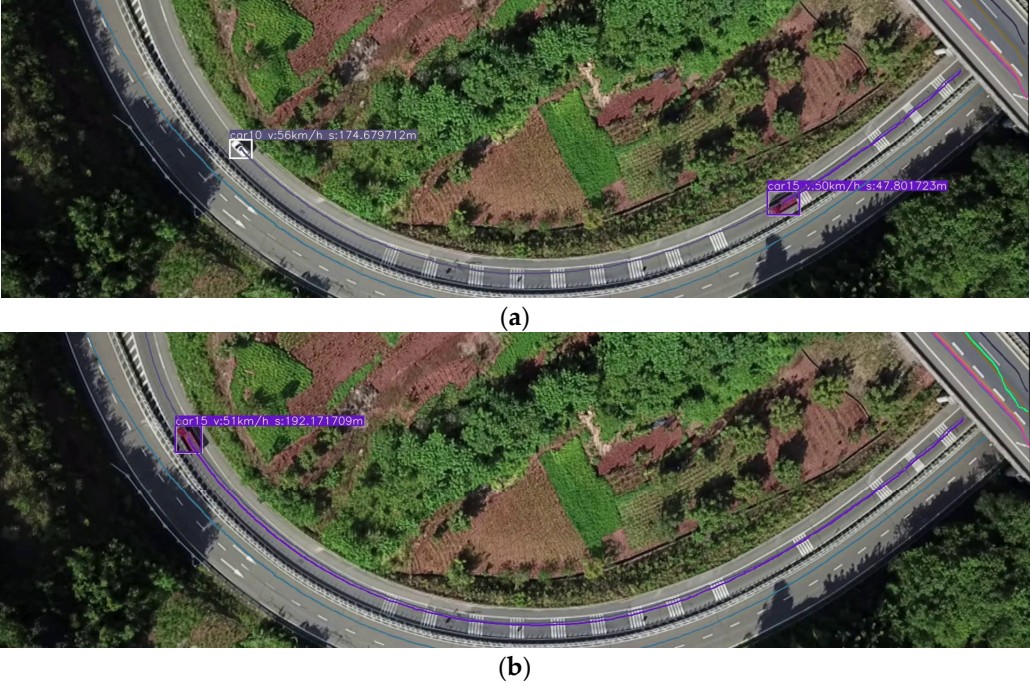

**Figure 5.** Registration effects: (**a**) trajectory of previous frame after registration; (**b**) trajectory of later frame after registration.

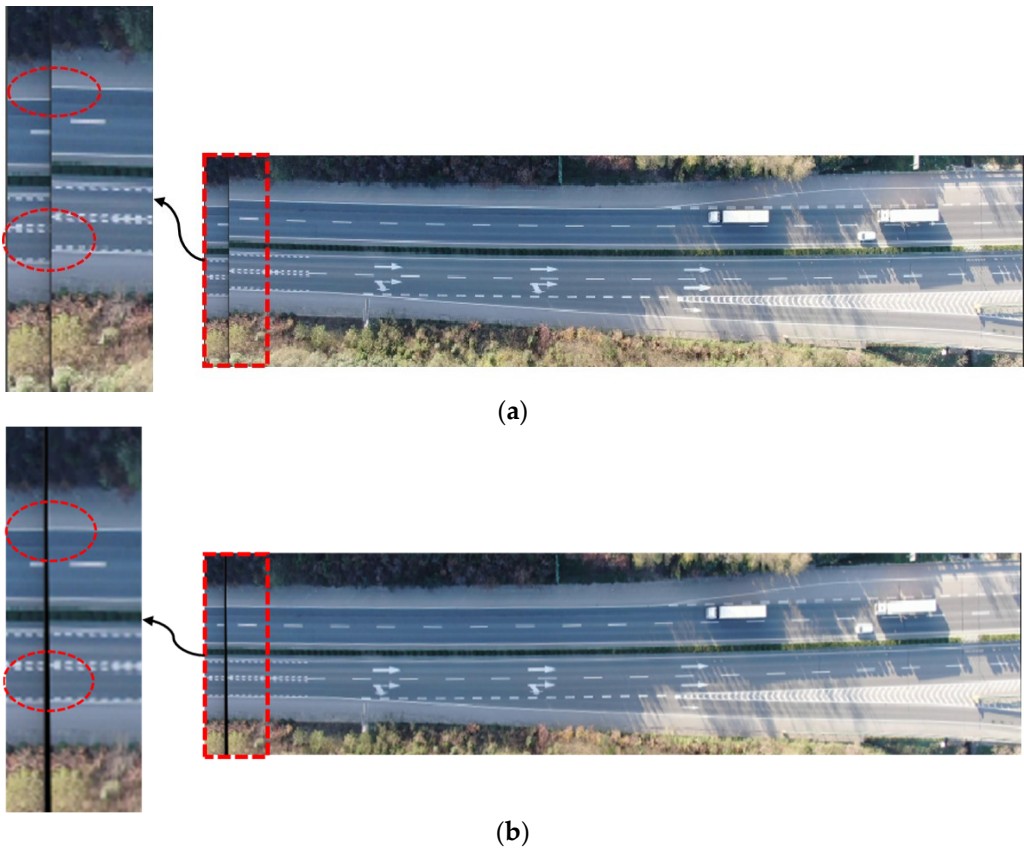

**Figure 6.** Comparison of different frames (**a**) before and (**b**) after registration.

The vehicle trajectory mistake in the unregistered video is primarily due to the shaking camera; despite our best efforts to use drone aerial photography in calm weather, it will still cause shaking of approximately the width of the lane, and the error is roughly 3~4 m. The trajectory error after registration is mainly generated by the oblique angle of view of aerial photography, which is generally about two pixels wide. When the height of the drone is 230 m, the real distance represented by one pixel is about 10 cm; therefore, by reprocessing the registered video, the trajectory error between the vehicle and the lane line can be controlled within 20 cm.

*4.2. Deep Learning-Based Vehicle Data Extraction*

4.2.1. Calibration of the Pixel Distance Parameter

Video registration is a preprocessing step that ensures data extraction accuracy. After registration, calibration must be conducted first, which involves calculating the ratio of the real distance to pixel distance. Drawing software such as Photoshop can be used to view the pixel coordinates of any image point. In this study, we used the 6 m dotted line in the road as the standard to calculate the real pixel distance equal to 1 m by calculating the pixel distance of the known length of the road, as shown in Figure 7. As the video is shot vertically, the image's horizontal 1/4 or 3/4 position is used for computation and calibration to minimize the viewing angle inaccuracy as much as possible.

$$k = \frac{\sqrt{(x_2 - x_1)^2 - (y_2 - y_1)^2}}{6} \tag{2}$$

where $k$ is the calibration parameter, $x_1$ and $y_1$ are the pixel coordinates of the first point, and $x_2$ and $y_2$ are the pixel coordinates of the second point. Here, it can be seen that the calibration parameter $k = 6.34$, that is, the pixel distance of 6.34 represents the real distance of 1 m. To avoid potential damage and blurring of a single marking line, length calibration

should be performed for at least 3 marking lines in good condition, and the average is used as the final distance calibration value.

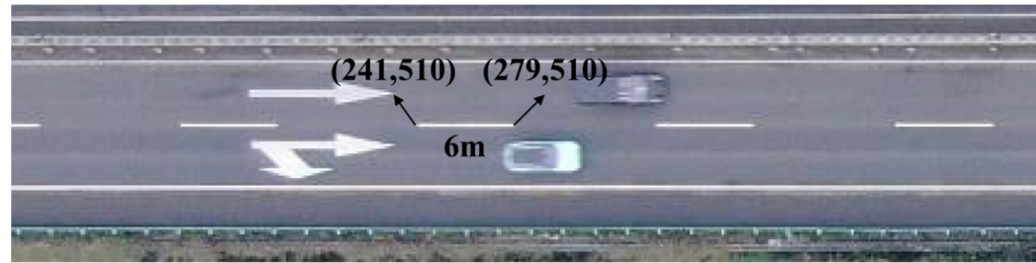

**Figure 7.** Distance parameter calibration.

4.2.2. Vehicle Detection Based on YOLOv5

YOLO's vehicle labeling and training are straightforward, and the vehicle frame selection is done on the training and analysis video, as illustrated in Figure 8. Model training was done with a ratio of training set: validation set: test set = 0.7:0.2:0.1, a total of 100 training rounds were completed, and the best test effect model was chosen for vehicle detection. After verification, the vehicle detection effect reached about 98%, and the detection frame was stable and not affected by shadows, as shown in Figure 9; however, there were also misidentifications, in which non-vehicle elements were incorrectly identified as vehicles. From the standpoint of data processing, subsequent solutions will be suggested. The results of the identification are shown in Table 4.

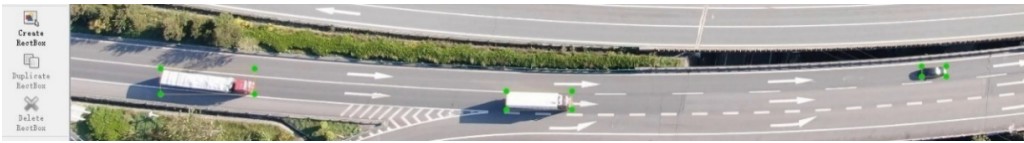

**Figure 8.** Labeling vehicles for training.

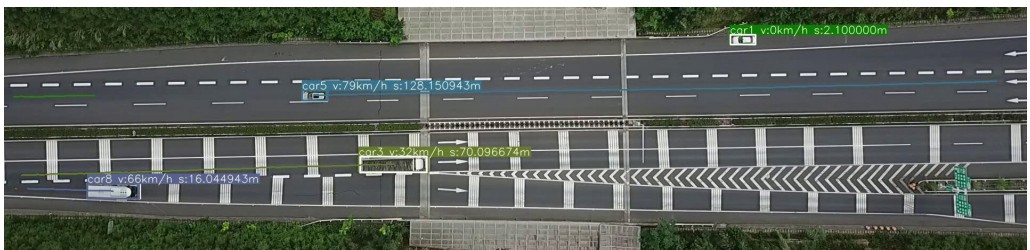

**Figure 9.** Result of vehicle detection.

**Table 4.** Result of vehicle detection.

| Road | Precision (%) | Recall (%) | True Negative | ID Switch (%) |
|---|---|---|---|---|
| Main section of Baomao expressway (road 1) | 95.00 | 98.53 | 3 | 5 |
| Tianquan interchange exit of Yakang expressway (road 2) | 92.35 | 97.60 | 7 | 12 |
| Duogong interchange ring ramp of Yakang expressway (road 3) | 95.00 | 96.00 | 4 | 5 |

The results of vehicle recognition and tracking, taking 3 different types of road sections as examples, were analyzed. As the training and application sets were from the same video,

the vehicle recognition and recall rates under diverse road circumstances could reach high values after adequate training.

The road markings and alignments on roads 1 and 2 were rather straightforward, and all indications attained a high level. ID switch mainly occurred during the process of vehicles entering the screen and could also occur due to occlusion between very few vehicles. All of the indicators on road 3 had lower values, and the ID switch was much higher than on other routes. The road markings on this part of the road are complicated, with relatively dense white horizontal markings and text, and when white vehicles interact with the complex noise elements, recognition accuracy suffers. At the same time, some non-vehicle objects in the image could be wrongly labeled as vehicles. A data cleaning method for this problem is proposed later.

### 4.2.3. Continuous Vehicle Tracking Using the DeepSORT Algorithm

The continuous tracking task must be done after utilizing YOLOv5 to identify cars in the image in difficult settings. The simple SORT algorithm with a Kalman filter and Hungarian algorithm as the core is widely used; however, SORT has many ID switches in some scenarios, such as occluded environments, resulting in poor tracking efficiency. Nicolai overcame this by using SORT to add appearance information, borrowing the re-identification (ReID) domain model to extract features, and reducing the number of ID switches, and proposed the DeepSORT algorithm, which is also the method used in this paper. Its main operation process is shown in Figure 10.

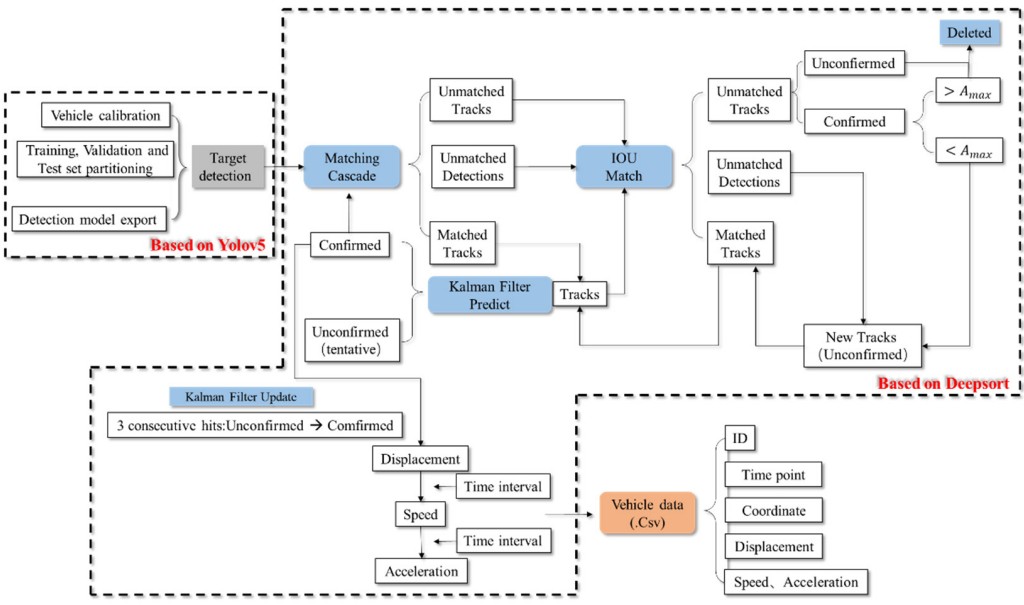

**Figure 10.** Continuous vehicle tracking process using DeepSORT algorithm.

After the detection and continuous tracking of the vehicle target are completed, as shown in Figure 11, the time interval and calibration parameters obtained in Section 4.2.1 can be used to collect information such as speed and acceleration.

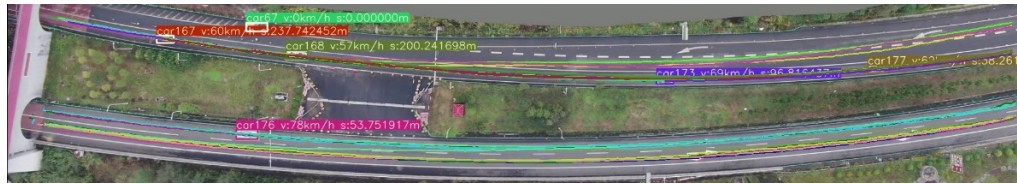

**Figure 11.** Continuous vehicle trajectory tracking.

#### 4.2.4. Validation of Data

It is vital to validate the accuracy of the vehicle speed recovered by the program in order to assure the accuracy of aerial video data extraction. We randomly selected 9 large and small cars from the data extracted from the exit section of the main line of Yakang Expressway and recorded the time it took to pass the dotted line on the road by playing the video frame by frame; as the real length of the dotted line on the road is known to be 6 m, the car's true speed can be determined. Then, the speed detection accuracy was compared with the speed extracted by machine vision.

$$v_r = \frac{n_{ic}}{n_0} \times 6 \times 3.6 \tag{3}$$

where $v_r$ is the real speed of the vehicle (km/h), $n_0$ is the video frame rate and $n_0 = 30$ in this paper, and $n_{ic}$ is the number of frames in which the $i$th vehicle passed the dotted line. The results of the data test are shown in Table 5.

**Table 5.** Data accuracy test.

| Type | Small Car | | | | | | | | | Large Car | | | | | | | | |
|---|---|---|---|---|---|---|---|---|---|---|---|---|---|---|---|---|---|---|
| Real speed (km/h) | 79 | 79 | 64.8 | 86 | 86 | 74.05 | 86.4 | 69 | 64.8 | 47.12 | 45 | 45 | 64.2 | 39.8 | 43.2 | 39 | 57.6 | 61.5 |
| Detected speed (km/h) | 77 | 79 | 67 | 80 | 82 | 79 | 91 | 66 | 68 | 45 | 41 | 40.5 | 65 | 36 | 39 | 39 | 59 | 60 |

The data accuracy test table of real and detected speed values shows that the speed accuracy is high. The highest error value and the overall accuracy were assessed to ensure that the error did not have a substantial influence on the real findings. Overall accuracy is characterized by mean accuracy (*T*) and root mean square error (*RMSE*):

$$T = \frac{\overline{v}_r}{\overline{v}} \times 100\% \tag{4}$$

where $T$ is the accuracy of average speed, $\overline{v}_r$ is the average value of the detected vehicle speed, and $\overline{v}$ is the average of real speed.

$$\text{RMSE} = \sqrt{\frac{1}{n}\sum_{t=1}^{n}\left(v_t{}' - v_t\right)^2} \tag{5}$$

where RMSE is the root mean square error of vehicle speed, $n$ is the number of cars, and $v_t{}'$ and $v_t$ are the real and detected speed of the $t$th vehicle.

As shown by the analysis results in Table 6, the maximum error values of large and small cars are not significantly different, and the maximum error rate of large cars is considerably higher. The maximum and minimum error values were deleted to reduce the overwhelming influence of data from specific circumstances on overall accuracy. The average accuracy of small cars is 98.5%, and the average accuracy of large cars is 95.7%. The RMSE of large cars is 2.94 km/h, and the RMSE of small cars is 3.74 km/h. The accuracy of the speed data extracted by deep learning is above 95% for large cars and 98% for small cars.

**Table 6.** Error analysis.

| Type | Max (Error) (km/h) | Maximum Error rate (%) | Mean Error (km/h) | Mean Error Rate (%) | T (%) | RMSE (km/h) |
|---|---|---|---|---|---|---|
| Large car | 5 | 10 | 2.98 | 6.06 | 95.7 | 2.94 |
| Small car | 6 | 6.98 | 3.89 | 5.08 | 98.5 | 3.74 |

### 4.3. Vehicle Data Associated with Lane Lines

#### 4.3.1. Lane Line Calibration

In this study, a method of pre-calibrating the lane line and then computing the distance is provided as a replacement for lane line identification, which can reliably obtain the relationship between vehicle speed and lane line distance. The benefits of this approach are that it is simple to use, has a consistent effect, and is not affected by the state of the lane lines. The procedure is as follows:

Take an image from an aerial video and import it into CAD software using the coordinate origin in the upper left corner as a guide. By scaling, the coordinate values of the image length and width are equal to the pixel length. The pixel coordinates of the image, which are the same as the vehicle *X* coordinates recovered in Section 4.2.3, correspond to the coordinates of any point in the figure in the CAD software in this situation, and the *Y* coordinates are opposite integers. Through the graphic design function of road design software (such as HintVR), the same design line as the lane line in the picture is fitted and designed, and then the pile-by-pile coordinates of the design line are output according to fixed intervals. The accuracy of the video environment in this paper can achieve coordinate output every 10 cm on a real road. Taking the negative value of the pile-by-pile coordinate table's *Y*-axis coordinate value, which is the lane line's pixel coordinate calibration file, its coordinates are in the same coordinate system as the vehicle coordinates identified and retrieved in Section 4.2 and can be calculated immediately. The lane line marking is shown in Figure 12.

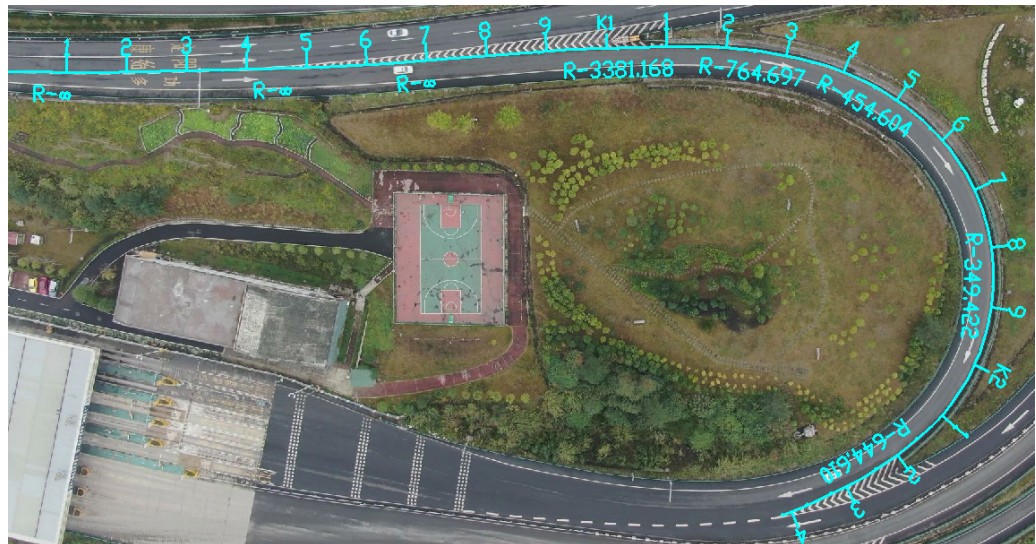

**Figure 12.** Lane marking.

#### 4.3.2. Section Speed Extraction

The continuous acquisition coordinates and related speed values of vehicles obtained in Section 4.2 are shown in Table 7, along with the lane line calibration file produced in Section 4.3.1.

The Python programming language was used to extract cross-section speed.

There are three layers of loops in all. To extract the continuous pixel coordinates and speed values of each car in the video, first traverse the numbers of all vehicles. Then, on the calibrated lane line, traverse each stake number to be extracted, collect every 1 m, and calculate the distance between each stake number and all coordinates of the numbered vehicles extracted from the previous layer.

$$l_k = \sqrt{[(s_{kx} - car\_x_{ij})^2 - (s_{ky} - car\_y_{ij})^2]} \tag{6}$$

where $l_k$ is the distance between stake $k$ and the $j$th collection point of the $i$th vehicle; $s_{kx}$ are the X- and Y-coordinates of the lane line at stake $k$, respectively; and $car\_x_{ij}$ are the X- and Y-coordinates of the $j$th collection point of the $i$th vehicle, respectively.

The two positions of the $i$th vehicle before and after stake $k$ are considered as the two minimum values of $l_k$. Between the two collecting stakes, the vehicle can be assumed to be driving at a constant speed. The speed $v_{ik}$ of the $i$th vehicle at stake $k$ is calculated using the front and rear speeds and the distance between the two points:

$$v_{ik} = v_2 + \frac{l_1}{l}(v_1 - v_2) \tag{7}$$

where $v_{ik}$ is the speed of the $i$th vehicle at stake $k$, $v_2$ is the vehicle speed at the collection point before stake $k$, $v_1$ is the vehicle speed at the collection point after stake $k$, $l_1$ is the the horizontal/vertical distance between stake $k$ and the collection point before stake $k$, and $l$ is the horizontal/vertical distance between the collection points before and after stake $k$.

**Table 7.** Data file.

| Time (s) | Vehicle ID | Speed (km/h) | Displacement | X | Y | Acceleration (m/s$^2$) |
|----------|-----------|--------------|--------------|------|------|------------------------|
| 91.09 | 10 | 45.00 | 2.72 | 1518.00 | 39.00 | 2.00 |
| 91.42 | 10 | 47.42 | 7.22 | 1511.00 | 75.00 | 2.01 |
| 91.76 | 10 | 48.29 | 11.55 | 1505.00 | 110.00 | 0.72 |
| 92.09 | 10 | 48.30 | 15.52 | 1499.00 | 142.00 | 0.01 |
| 92.43 | 10 | 47.10 | 19.50 | 1493.00 | 174.00 | −1.00 |
| | | | ... ... | | | |

Note: X and Y coordinates shown in the table are pixel coordinates.

The speed of each vehicle in each section can be determined by using this procedure to traverse each vehicle and section. Similarly, as indicated in Table 8, information such as vehicle speed and acceleration for all cars at each stake can be retrieved.

**Table 8.** Speed data file.

| Real Stake | Pixel Stake | Car 2 Speed | Car 2 Acceleration | Car 3 Speed | Car 3 Acceleration | Car 9 Speed | Car 9 Acceleration |
|-----------|-------------|-------------|--------------------|-------------|--------------------|-------------|--------------------|
| 0 | 0 | 102.535 | 6.696 | 99.271 | −0.641 | 91.397 | 0.081 |
| 5 | 40 | 102.535 | 6.696 | 100.281 | 0.842 | 92.816 | 1.183 |
| 10 | 80 | 102.444 | −0.0759 | 100.281 | 0.842 | 92.816 | 1.183 |
| 15 | 120 | 102.444 | −0.0759 | 100.435 | 0.128 | 92.860 | 0.037 |
| 20 | 160 | 102.911 | 0.389 | 100.435 | 0.128 | 92.860 | 0.037 |
| | | | ... ... | | | | |

### 4.3.3. Extraction of Lane Line and Lateral Vehicle Distance

Through the method described in Section 4.3.2, distance $l_k$ between stake $k$ and the $j$th collection point of the $i$th vehicle can be obtained. We take the minimum two values of $l_k$, $l_{k1}$, and $l_{k2}$, which are the distances between the two positions of the $i$th vehicle before and after stake $k$, as shown in Figure 13.

If the coordinates of the three locations in the triangle created by the two collection points and stake $k$ are known, the lengths of the triangle's three sides can be calculated. The height $h$ of this triangle is the lateral distance of the car at stake $k$. Heron's formula can be used to calculate the vertical distance $h$ of each vehicle from the lane line at each stake:

$$h = \frac{2}{d}\sqrt{p(p-d)(p-l_{k1})(p-l_{k1})} \tag{8}$$

where $h$ is the distance between stake $k$ and the car; $d$ is the length between collection points 1 and 2 and is a known quantity, $d = \sqrt{(x_1 - x_2)^2 + (y_1 - y_2)^2}$; and $x_1$, $x_2$, $y_1$, $y_2$ are the X-

and *Y*-coordinates of collection points 1 and 2, respectively. $p = \dfrac{l_{k1} + l_{k2} + d}{2}$ represents half the circumference.

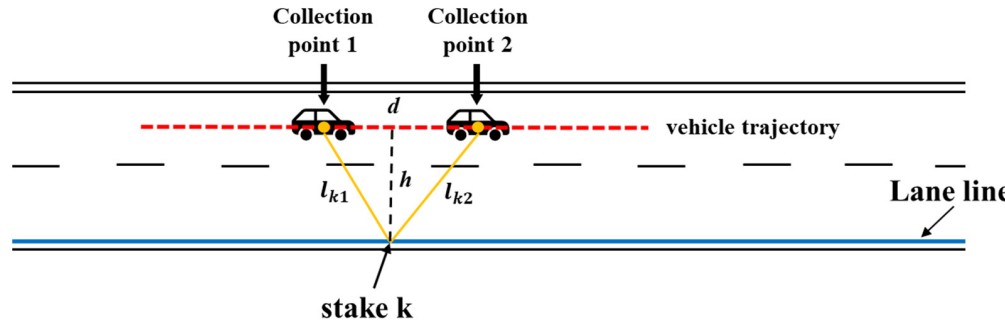

**Figure 13.** Schematic diagram of lateral distance extraction.

This method can be used at each lane line stake to determine the lateral distance between each section of the lane line and each vehicle, which is the vehicle's offset. The lateral speed and lateral acceleration of the vehicle relative to the lane line can be calculated using the time difference. Table 9 shows the complete data composition.

**Table 9.** Data file for vehicle detection.

| Real Stake | Pixel Stake | Car 2 Speed | Car 2 Acceleration | Car 2 Lateral Distance | Car 2 Lateral Speed | Car 2 Lateral Acceleration | Car 3 Speed |
|---|---|---|---|---|---|---|---|
| 0 | 0 | 102.535 | 6.696 | 1.962 | −0.274 | −0.175 | … … … |
| 5 | 40 | 102.535 | 6.696 | 1.908 | −0.274 | −0.175 | … … … |
| 10 | 80 | 102.444 | −0.0759 | 1.854 | −0.332 | −0.175 | … … … |
| 15 | 120 | 102.444 | −0.0759 | 1.814 | −0.332 | −0.175 | … … … |
| 20 | 160 | 102.911 | 0.389 | 1.826 | −0.021 | 0.933 | … … … |
| | | | | … … | | | |

In Table 9, Car 2 indicates the vehicle ID, and the longitudinal direction is the lane line stake, including the converted real distance stake and pixel coordinate stake. The lateral data include vehicle speed, acceleration, lateral distance, lateral velocity, and lateral acceleration information for each vehicle. The columns represent the vehicles' continuous data, and the rows represent the speed and distance information for all cars at the stake. According to the vehicle number, collection coordinates, and collection time point, data such as the distance and the speed differential between adjacent vehicles can be collected in the follow-up.

### 4.3.4. Data Cleaning

After many aerial video vehicles have been trained, the vehicle recognition accuracy can be guaranteed to be 100%; however, when confronted with a new video after additional verification, the following scenarios may arise:

1.  Vehicle misidentification can occur, including the possibility of mistaking non-vehicle elements for vehicles, although this is uncommon.
2.  A vehicle is temporarily blocked by a gantry or sign, resulting in intermittent recognition.
3.  Data from vehicles outside the lane being studied will be collected.
4.  When the vehicle first appears on the screen, there is a gradual recognition process. As the vehicle's middle is chosen as the detecting point, the speed information entered onto the screen may be erroneous.
5.  A few vehicles may have number switching.

The following solutions and data cleaning procedures are used to address these identification issues and noisy data:

1. As the incorrectly identified items are all fixed objects on the screen, all data with an identification speed of less than 5 km/h are exported to 0 km/h in the software, which can be erased directly thereafter.
2. As long as the vehicle number does not change during the recognition interruption (caused by the obstruction of the gantry, sign, etc.) for around one second, there is no problem with the data.
3. The distance between all vehicles and the calibration lane line can be calculated; the distance greater than the width of the lane is the vehicle data of the remaining lanes, and all data of its number can be deleted.
4. The problem of gradual recognition of vehicles entering the screen is unavoidable. The collection range can be appropriately larger than the road section to be studied, then data within 10 m of the entry process can be deleted.
5. The number switching of a small number of vehicles can correspond to the numbering unification; when more vehicles are switched, the problem can be solved by increasing the data training.

Following the data cleaning procedure outlined above, high-quality, high-precision continuous vehicle speed and trajectory data can be obtained.

### 4.3.5. Frenet Coordinate System

The Cartesian coordinate system is usually used to describe the position of objects; however, it is not the greatest choice for cars traveling on curving roadways. Currently, vehicle coordinate information is in the form of pixel coordinates, and the origin of the Cartesian coordinate system is in the upper left corner of the image. When driving on a road with variable alignment, determining the position of the vehicle on the road might be challenging. The Frenet coordinate system describes the position of the car relative to the road, which can better describe the lane lines and the state of the vehicle on the road. In the Frenet coordinate system, $s$ is the ordinate, representing the distance along the road, and $d$, the abscissa, is the distance from the longitudinal line, as shown in Figure 14. Traditionally, vehicle speed and trajectory data are stored in the Cartesian coordinate system and converting to the Frenet coordinate system is a time-consuming process. With the help of the Frenet coordinate system, the data distribution of vehicles can be easily established using vehicle and lane line contrasting techniques.

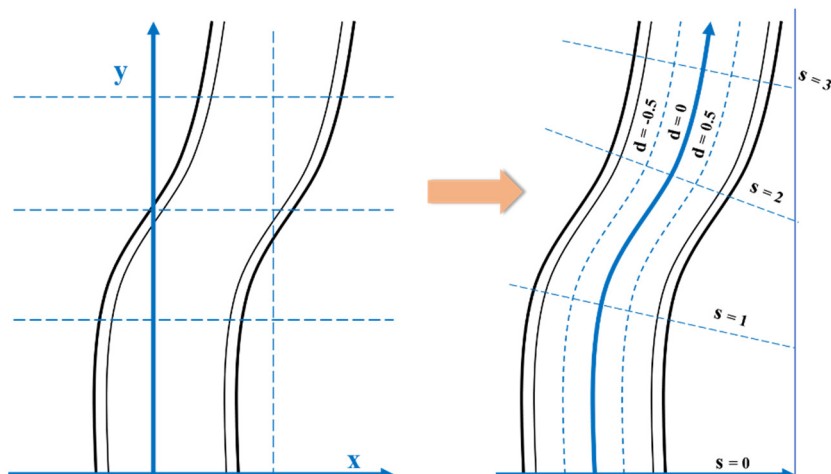

**Figure 14.** Converting Cartesian coordinate system to the Frenet coordinate system.

### 5. Results

We drew the data of the investigated road sections, and each road section included three parts: aerial image, trajectory data, and vehicle speed data. Using the data, we can see the acceleration and deceleration behavior of each road section, as well as the changes

in distance between the vehicle and the lane line at various points. As seen in Figure 15, there is a degree of consistency in driving behavior.

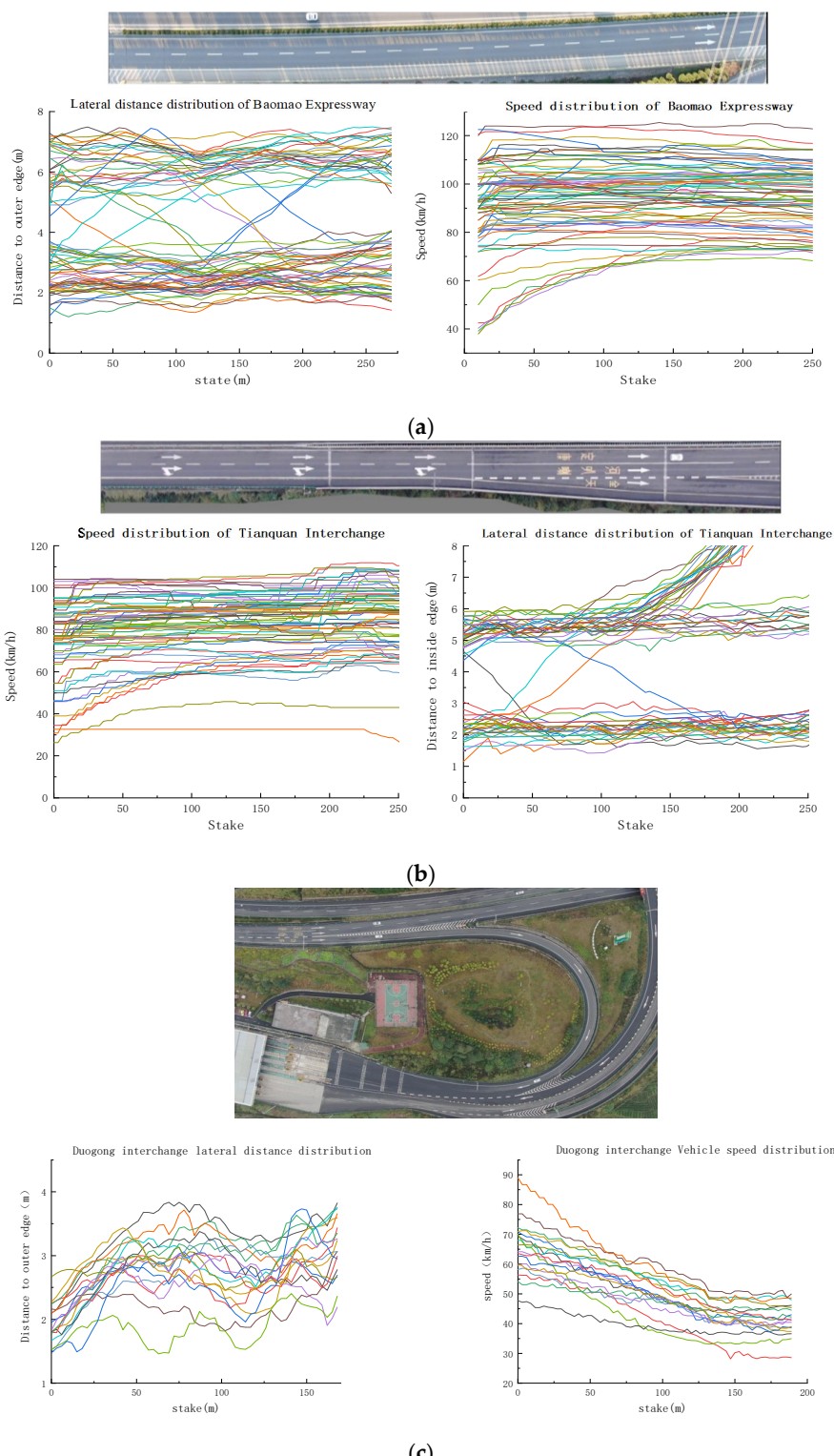

**Figure 15.** *Cont.*

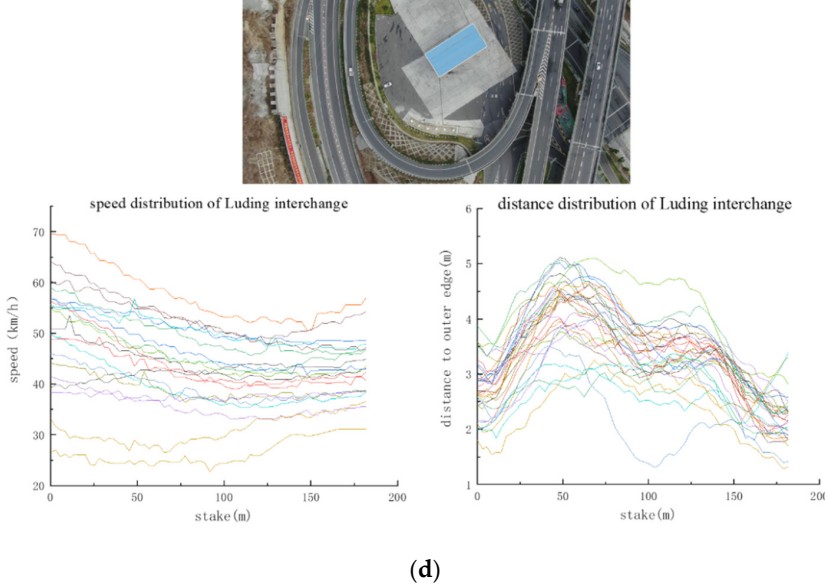

(**d**)

**Figure 15.** Data of collected road sections: trajectory and speed data of (**a**) main line of Baomao Expressway; (**b**) Tianquan interchange; (**c**) Duogong interchange ramp; (**d**) Luding interchange ramp.

The entire road segment can be divided into numerous squares of the same size using the Frenet coordinate system, depending on the distance between the d-axis and the s-axis. The average speed distribution of the entire road segment can be derived by computing the average vehicle speed of all trajectories passing through this location. The average vehicle speed graph clearly shows the impact of the interchange exit on the main line vehicle speed, as shown in Figure 16.

$$v_{ij} = \frac{\sum\limits_{k=1}^{n} v_{ijk}}{n} \tag{9}$$

where $v_{ij}$ is the average vehicle speed at the position in row $i$ and column $j$, $v_{ijk}$ is the speed of the $k$th passing vehicle at the position of the $i$th row and $j$th column, and $\sum\limits_{k=1}^{n} v_{ijk}$ indicates the sum of $n$ vehicle speeds; $n$ refers to the total number of vehicles that passed the position of the $i$th row and $j$th column.

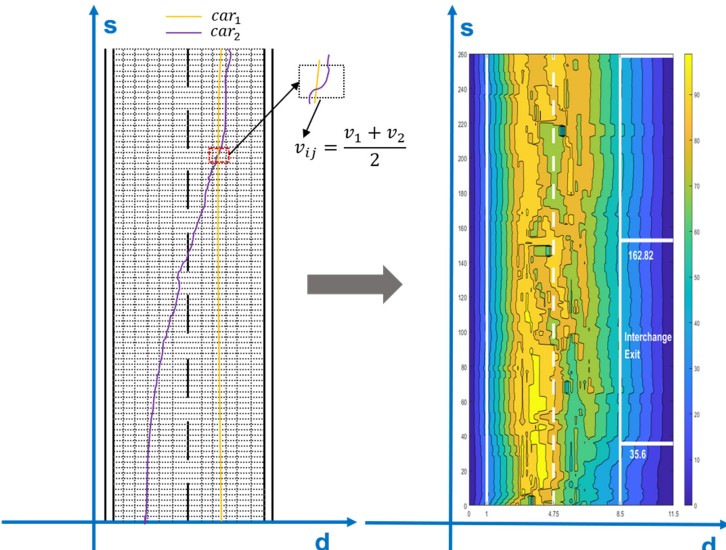

**Figure 16.** Average speed of vehicles at the exit of the Baomao Expressway.

## 6. Discussion and Conclusions

In this study, we formulated a more reliable method for extracting freeway vehicle data. This method utilizes the SIFT, YOLOv5, and DeepSORT algorithms to counteract the vibration of UAVs and realize whole-domain vehicle recognition and continuous tracking. An algorithm for lane line calibration and cross-section information acquisition was also proposed. Compared with current methods, our method effectively improves vehicle speed and trajectory data by eliminating the important but rarely mentioned video shaking problem and improves vehicle trajectory tracking accuracy from 3.54 to 30 cm. The problems of the poor stability and inaccuracy of current image-based lane lines, which cannot meet the needs of actual situations, are solved by lane line calibration, which also addresses the stable association between vehicle data and lane lines, and the structure of vehicle speed and trajectory data, which is beneficial to the establishment of the Frenet coordinate system.

Although there has been progress in video-based target recognition and tracking with the advent of deep learning technology (and this work presents a more reliable data extraction approach), there are still some issues that need to be addressed in the future, such as the following:

(1) Accurate recognition of small target vehicles is still a challenge when using a larger range of aerial video.
(2) Target recognition training using deep learning is still a somewhat hard task, which we can look into simplifying.
(3) The data gained through video-based target recognition are not entirely usable. Data filtering was carried out in this work, but a more effective way to evaluate and optimize speed and trajectory data remains a key research focus.

**Author Contributions:** C.Z., Z.T. and B.W. developed the method and designed the experiments; Z.T. performed the experiments; M.Z. and B.W. analyzed the results; L.H. contributed to the use of analysis tools; all authors contributed to writing and reviewing the manuscript. All authors have read and agreed to the published version of the manuscript.

**Funding:** This research was funded by Chang'an University (Xi'an, China) through the National Key Research and Development Program of China (grant numbers 2020YFC1512005, 2020YFC1512002) and Sichuan Science and Technology Program (NO:2022YFG0048).

**Institutional Review Board Statement:** Not available.

**Informed Consent Statement:** Not available.

**Data Availability Statement:** Not available.

**Acknowledgments:** The authors would like to thank the Communication Surveying and Design Institute for providing us with the road design data.

**Conflicts of Interest:** The authors declare no conflict of interest.

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
