# Peer review of "Developing a More Reliable Aerial Photography-Based Method for Acquiring Freeway Traffic Data"

_remotesensing, doi:10.3390/rs14092202_

Round 1
Reviewer 1 Report
Dear authors,
The paper is interesting and in general well written. I provided comments directly applied to the pdf file in Acrobad Reader tool.

Author Response
Dear reviewer:
Please see the attachment. I have provided responses to all comments in the PFD file.

Reviewer 2 Report
In this paper the authors conduct a study to formulate a more reliable method for real traffic data acquisition, which outperforms the traditional methods in terms of data accuracy and integrity.
The paper is well structured, and their different sections are correctly explained, however, to improve the quality of the article, I consider that the following improvements should be made
I believe that a section should be included where the entire proposed process is described in general, this will give the reader a better overview and understanding of your proposal. This section can be complemented with a diagram that represents the phases that make up the proposal.
A more in-depth analysis of the results obtained in Table 5 is required.
The results section is very weak. A more in-depth analysis of the results obtained is needed. This section is where they should show how solid their proposal is, showing the benefits it has over other proposals analyzed.
Author Response
Response to Reviewer 2 Comments
Point 1: I believe that a section should be included where the entire proposed process is described in general, this will give the reader a better overview and understanding of your proposal. This section can be complemented with a diagram that represents the phases that make up the proposal.
Response 1: In Chapter 4, Figure 3, I described the whole process, but the description is not sufficient, so I supplemented this part of the text in my new version to give the reader a better overview and understanding of my proposal.
Point 2: A more in-depth analysis of the results obtained in Table 5 is required.
Response 2: OK,I have enriched this part in my new version as following:
Taking three different forms of road sections as an example, the result of vehicle recognition and tracking is analyzed. Because the training set and the application set are the same video, the vehicle's recognition rate and recall rate under diverse road circum-stances can reach a high value after adequate training.
The road markings and alignments on Road 1 and Road 2 are rather straightfor-ward, and all indications have attained a high level. ID switch mainly occurs in the pro-cess of vehicles entering the screen, and occlusion between very few vehicles may can al-so cause ID switch. All of the indicators on road 3 are lower, and the ID switch is much higher than on other routes. Because the road markings on this part of road are compli-cated, and there are relatively dense white horizontal markings and texts on the road, when white vehicles collide with the complex noise elements, recognition accuracy suffers. Simultaneously, some non-vehicle objects in the image could be wrongly labeled as vehicles. This paper also proposes a data cleaning method for this problem later.
Point 3: The results section is very weak. A more in-depth analysis of the results obtained is needed. This section is where they should show how solid their proposal is, showing the benefits it has over other proposals analyzed.
Response 3: In my new version, I changed my “Conclusion” to “Discussion and conclusion”, and added the following:
Compared with current methods,our method effectively improves vehicle speed and tra-jectory data by eliminating the important but rarely mentioned video shaking problem, and improves vehicle trajectory tracking accuracy from 3.54m to 30cm. The problems of poor stability and accuracy of current image-based lane lines, which cannot meet actual needs, are solved by lane line calibration, as is the stable association between vehicle data and lane lines, and the structure of vehicle speed and trajectory data, which is beneficial to the establishment of the Frenet coordinate system.
Although video-based target recognition and tracking has progressed with the advent of deep learning technology, and this work also presents a more reliable data extraction approach, there are still some issues that need to be addressed in the future, such as: 1. Accurate recognition of small target vehicles is still a challenge when using a larger range of aerial video; 2. Target recognition training using deep learning is still a somewhat hard task, which we can look into simplifying in the future. 3. The data gained through vid-eo-based target recognition is not entirely usable.Data filtering has been done in this work, but a more effective way for evaluating and optimizing speed and trajectory data remains a key research focus.

Reviewer 3 Report
The paper stands as an interesting piece of research with solid methodology and a good description of the mathematical background of the proposed method. Validation is also assessed as a plus in the overall investigation. Its content suits the journal aim and the paper can be reconsidered after some revisions as per the following suggestions for further improvement:
(1) To the reviewer’s opinion the issue of road safety is a social and imperative demand that depends on two main pillar factors: (a) driving behavior and features and (b) roadway infrastructure condition (road geometry and most importantly pavement surface condition – skid resistance and polishing material behavior). The authors emphasized on the first scale. However, vehicle speed depends on the pavement surface condition and many models link surface friction and vehicle speed in the framework of road safety research and tire-pavement interaction modeling. Suggested literature to be included in the paper: https://doi.org/10.1016/j.ijtst.2021.09.004 , https://doi.org/10.1016/j.conbuildmat.2016.04.002 . I do believe the authors need to make this important clarification at the beginning of the introduction or at the beginning of section 2. For instance, at the latter point it is stated that “Road safety assessment can be conducted using different techniques…”, but again the role of pavement infrastructure is missing. So, a clarification of how the authors perceive road safety is recommended.
(2) The objective is not clear. If it set in lines 62-64/130-131, then the authors need to revise it and better demonstrate it.
(3) Line 133: A terminology issue is again suggested. “Road profile” is in general perceived as the pavement surface irregularities (known as pavement roughness). The authors might mean different types of road geometry aspects (i.e. straight roads, sharp turns), so it is recommended to refine.
(4) From the presented photos, the reviewer thinks that the proposed method is applicable for highway motorways where high speeds are usually met. Does it provide reliable estimations for urban roadways where traffic incidents are more usual? Please elaborate on this practice-oriented aspect.
(5) A discussion section of the presented results is missing and conclusions are very short. Please add any limitations of your method and aspects for further improvements in your methodology.
(6) I suggest using quotation marks at terms like “You Only Look Once” in the abstract and wherever else used.
(7) Please make a necessary language check for minor typos.
Author Response
Response to Reviewer 3 Comments
Point 1: To the reviewer’s opinion the issue of road safety is a social and imperative demand that depends on two main pillar factors: (a) driving behavior and features and (b) roadway infrastructure condition (road geometry and most importantly pavement surface condition – skid resistance and polishing material behavior). The authors emphasized on the first scale. However, vehicle speed depends on the pavement surface condition and many models link surface friction and vehicle speed in the framework of road safety research and tire-pavement interaction modeling. Suggested literature to be included in the paper: https://doi.org/10.1016/j.ijtst.2021.09.004 , https://doi.org/10.1016/j.conbuildmat.2016.04.002 . I do believe the authors need to make this important clarification at the beginning of the introduction or at the beginning of section 2. For instance, at the latter point it is stated that “Road safety assessment can be conducted using different techniques…”, but again the role of pavement infrastructure is missing. So, a clarification of how the authors perceive road safety is recommended.
Response 1: Thanks, I think “roads will have a variety of effects on traffic safety. Aside from the road alignment, which affects the driver's judgment, the markings, pavement materials, and damage all have an impact on the vehicle's driving condition. However, these factors are ultimately described by the vehicle speed and trajectory, so based on the vehicle speed and trajectory is a comprehensive road safety evaluation.”
And thanks for your recommendation, I have add https://doi.org/10.1016/j.ijtst.2021.09.004 , https://doi.org/10.1016/j.conbuildmat.2016.04.002 in my paper.
Point 2: The objective is not clear. If it set in lines 62-64/130-131, then the authors need to revise it and better demonstrate it.
Response 2: The objective set in lines 62-64, The content of lines 130-131 is inappropriate and may confuse the reader, so I have revised it in my new version.
Point 3: Line 133: A terminology issue is again suggested. “Road profile” is in general perceived as the pavement surface irregularities (known as pavement roughness). The authors might mean different types of road geometry aspects (i.e. straight roads, sharp turns), so it is recommended to refine.
Response 3: Thanks, I have corrected “road profile” to “road alignment”.
Point 4: From the presented photos, the reviewer thinks that the proposed method is applicable for highway motorways where high speeds are usually met. Does it provide reliable estimations for urban roadways where traffic incidents are more usual? Please elaborate on this practice-oriented aspect.
Response 4: Because the focus of my research is on highway traffic safety, so all of the scenarios depicted are highway scenes. We can see from our additional practice that this method is still adequate for slow-speed urban roadways, but it does not perform well in small-volume and high-speed target identification. However, it does not appear in actual road vehicle detection.
Point 5: A discussion section of the presented results is missing and conclusions are very short. Please add any limitations of your method and aspects for further improvements in your methodology.
Response 5: Thanks, I have enriched my “Conclusion” to “Discussion and conclusion” in my new version.
Point 6: I suggest using quotation marks at terms like “You Only Look Once” in the abstract and wherever else used.
Response 6: Thanks, I have corrected it in my new version.
Point 7: Please make a necessary language check for minor typos.
Response 7: Thanks, I will find a professional English editing agency to polish the language .

Round 2
Reviewer 3 Report
The authors have improved the manuscript and my comments were addressed. Thank you. I have no further comments, except these two:
- Lines 554-558: You might put these arguments in bullet points (separate paragraphs).
- Reference list: To avoid delays in the production phase, you might provide the DOI for those items not currently included in a specific volume (e.g. No. 15 and 34). Also, delete No. 52 if it does not correspond to a reference.
Author Response
Response to Reviewer 3 Comments
Point 1: Lines 554-558: You might put these arguments in bullet points (separate paragraphs).
Response 1: Thanks, I have corrected it in my new version.
Point 2:Reference list: To avoid delays in the production phase, you might provide the DOI for those items not currently included in a specific volume (e.g. No. 15 and 34). Also, delete No. 52 if it does not correspond to a reference.
Response 2: Thanks, I have checked my reference. And, I have added the volume information (e.g. No.34) and DOI information (e.g. No.15).